# My Mind is Working Overtime—Towards an Integrative Perspective of Psychological Detachment, Work-Related Rumination, and Work Reflection

**DOI:** 10.3390/ijerph16162987

**Published:** 2019-08-20

**Authors:** Oliver Weigelt, Petra Gierer, Christine J. Syrek

**Affiliations:** 1Organizational and Personnel Psychology, University of Rostock, D-18051 Rostock, Germany; 2Work and Organizational Psychology, University of Hagen, University of Hagen, D-58084 Hagen, Germany; 3Business Psychology, University of Applied Sciences Bonn-Rhein-Sieg, Von-Liebig-Str. 20, D-53359 Rheinbach, Germany

**Keywords:** rumination, psychological detachment, perseverative cognition, work reflection, vitality, burnout, thriving, work engagement, employee well-being, mental health

## Abstract

In the literature on occupational stress and recovery from work, several facets of thinking about work during off-job time have been conceptualized. However, research on the focal concepts is currently rather diffuse. In this study we take a closer look at the five most well-established concepts: (1) psychological detachment, (2) affective rumination, (3) problem-solving pondering, (4) positive work reflection, and (5) negative work reflection. More specifically, we scrutinized (1) whether the five facets of work-related rumination are empirically distinct, (2) whether they yield differential associations with different facets of employee well-being (burnout, work engagement, thriving, satisfaction with life, and flourishing), and (3) to what extent the five facets can be distinguished from and relate to conceptually similar constructs, such as irritation, worry, and neuroticism. We applied structural equation modeling techniques to cross-sectional survey data from 474 employees. Our results provide evidence for (1) five correlated, yet empirically distinct facets of work-related rumination. (2) Each facet yields a unique pattern of association with the eight aspects of employee well-being. For instance, detachment is strongly linked to satisfaction with life and flourishing. Affective rumination is linked particularly to burnout. Problem-solving pondering and positive work reflection yield the strongest links to work engagement. (3) The five facets of work-related rumination are distinct from related concepts, although there is a high overlap between (lower levels of) psychological detachment and cognitive irritation. Our study contributes to clarifying the structure of work-related rumination and extends the nomological network around different types of thinking about work during off-job time and employee well-being.

## 1. Introduction

There is a large body of research in occupational health psychology and other fields that suggests that unwinding after work during off-job time is important to avoid health-impairing effects [1]. Several approaches in occupational stress research assume that work-related rumination prolongs stress-related affective and physiological activation and hence contributes to impairing somatic health in the long run [2,3]. For instance, the concept of psychological detachment or sense of “being away from the work situation” [4,5] (p. 579) has been studied extensively over the last two decades yielding hundreds of empirical studies [6,7]. However, Cropley and colleagues [8] argue that lack of detachment may be one, but not the only, relevant aspect of thinking about work in leisure time. More specifically, besides (lack of) psychological detachment, they have distinguished two additional aspects of thinking about work: affective rumination and problem-solving pondering [9]. Affective rumination refers to pervasive, recurrent negative thoughts about work. By contrast, problem-solving pondering implies prolonged thinking about improving one’s performance or solving work-related problems—an activity that is less intrusive and may even be enjoyable [9]. Furthermore, there is research on yet another form of thinking about work: positive work reflection [10]. Positive work reflection encompasses thinking about the good sides of one’s job and realizing what one likes about one’s job [10,11]. Finally, a couple of studies have also considered negative work reflection, that is, thinking about the bad sides of one’s job [11,12]. In this paper, we take a broad perspective on different forms of work-related rumination.

We have reviewed the literature on work-related rumination. From our literature search psychological detachment, affective rumination, problem-solving pondering, positive work reflection, and negative work reflection are the five major aspects (and scales) studied in mainstream (top-tier) industrial and organizational psychology and occupational health psychology journals. We supplement our analysis by concomitants of work-related rumination like, for instance, irritation. Our focus is also in line with recent reviews and meta-analyses on psychological detachment and recovery [6,7]. Our manuscript is aimed at informing researchers interested in the established facets of work-related rumination. We believe that the list of facets is not arbitrary but exhaustive (regarding work-related rumination), although some authors may use varying labels for essentially similar aspects of work-related rumination [13]. We believe that the five aspects of thinking about work in leisure time have considerable overlap in content and we propose that they should be studied in a more integrated way. Comparison of the five facets is important because most researchers have studied each facet of thinking about work only in isolation. In our view, this state of affairs is unfortunate, because considering psychological detachment, affective rumination, problem-solving pondering, and work reflection in a disconnected way is not conducive to the development of a coherent and integrative theory of work-related rumination. Moreover, authors from different labs also tend to use similar arguments regarding rumination but apply different operationalizations/measures of rumination. At this point, we see a need for clarification whether the different facets/scales measure something different or whether essentially the same phenomenon is given just different labels by different people (e.g., affective rumination vs. negative work reflection). Given that the research on positive and negative work reflection is disconnected from the other conceptualizations of thinking about work, we lack information on whether researchers are actually studying distinct facets. For instance, if negative work reflection is basically duplicating affective rumination, merging the two concepts into one would be warranted. By contrast, if positive work reflection goes beyond lack of detachment or problem-solving pondering in predicting employee well-being, then adding positive reflection to the tripartite conceptualization of work-related rumination described above [9] would truly enhance our knowledge of thinking about work during off-job time. Beyond merely distinguishing facets gaining a comprehensive picture of the associations among the facets is important when having to decide for or against a specific operationalization.

Accordingly, in this study we take an integrative perspective and aim to consider the five major concepts of thinking about work from the literature within one study. We follow Martin and Tesser’s definition of rumination as a “class of conscious thoughts that revolve around a common instrumental theme and that recur in the absence of immediate environmental demands requiring the thoughts” [14] (p. 7). This definition is not confined to negative forms of thinking (about work) [13] and allows integration of positive work reflection and problem-solving pondering under the general umbrella of work-related rumination. We apply structural equation modeling techniques to factor analyze the focal scales using a comprehensive set of variables from a cross-sectional survey. Furthermore, we examine to what extent the five facets of thinking about work differentially relate to and incrementally predict several aspects of hedonic and eudemonic well-being. In this sense, our study aids researchers interested in the convergent and discriminant validity of the five facets of work-related rumination. More specifically, our study examines the extent to which the different facets of work-related rumination add up incrementally to explain variance in different aspects of employee well-being. Our emphasis is on differential associations. Hence, we address questions like “Which facet of work-related rumination relates particularly strongly and uniquely to which facet of well-being?” We consider aspects of work-related as well as general well-being. More specifically, we examine links to burnout, work engagement, and thriving, representing aspects of work-related well-being and satisfaction with life, and flourishing, representing aspects of general (non-context-specific) well-being. We explicitly consider a broad set of well-being outcomes in a cross-sectional survey. The breadth of concepts is a major strength of this study, which we would have to give up, for instance, in a longitudinal study. Although, bivariate links between work-related rumination and well-being may be subject to method bias, differential links to well-being (convergent and discriminant validity) are not trivial and cannot be explained easily by methodological artifacts. Predicting time lagged well-being might even provide less clear conclusions, if the five facets of rumination differ regarding the speed in which they affect a given indicator of employee well-being. If, for instance, affective rumination yields immediate effects, but lack of detachment affects well-being at a slower speed and in the long run only, comparing the effect of affective rumination vs. psychological detachment on burnout at t2 (e.g., two months or one year later) would be ambiguous [15,16]. In general terms, our study is meant to be a first step of providing a clearer picture of the nomological network around work-related rumination and related concepts. Notably, we consider a couple of correlates which have not been linked to the five facets of work-related rumination. Within this broad perspective, our study offers insights into the function of positive and negative work reflection with regard to employee well-being vis-à-vis the more established facets from the tripartite conceptualization of work-related rumination. Ultimately, our results will allow researchers to make informed decisions on choosing constructs that fit the aims and scope of their studies on thinking about work.

### 1.1. Switching Off—A Brief Review of the Research on Psychological Detachment

Among the facets of work-related rumination described above, psychological detachment has received the most attention in terms of empirical research so far. Sonnentag and Bayer [17] have described psychological detachment in terms of “not thinking about one’s work during off-job time” (p. 393). In their narrative review, Sonnentag and Fritz [18] conceptualize detachment as a psychological variable, which either mediates stressor–strain relationships or attenuates detrimental links between job stressors and well-being or performance outcomes. A recent meta-analysis [6] suggests that detachment yields the strongest negative associations with burnout and fatigue. Furthermore, detachment is positively related to a number of well-being outcomes, such as satisfaction with life, state of recovery, and affective well-being. Interestingly, however, psychological detachment is to some extent ambiguous with regard to the content and the affective valence of work-related thoughts [6,11]. For instance, low levels of detachment may arise equally from reconsidering an interesting work-related question or from nervous thoughts about a recent episode of conflict at work.

### 1.2. Beyond Psychological Detachment—Affective Rumination and Problem-Solving Pondering

Given the conceptual ambiguity of psychological detachment, Cropley and Zijlstra [9] have made a case for broadening the perspective to other forms of thinking about work beyond psychological detachment. More specifically, they propose considering two additional forms of thinking about work besides psychological detachment, namely, affective rumination and problem-solving pondering. Affective rumination has been described in terms of intrusive, pervasive, recurring thoughts about work, which are experienced as negative in affective terms [9]. In other words, affective rumination refers to negative emotional experiences triggered by job-related thoughts after work [8]. By contrast, problem-solving pondering refers to merely reconsidering work-related issues during free time [8]—an activity that does not involve the emotional process that sustains arousal as in affective rumination [19]. More specifically, Cropley and Zijlstra [9] argue that “people may ponder work-related problems because they find the act of thinking about work interesting. In more general terms this suggests that when people keep thinking about their work because they enjoy their work, and it helps them to solve issues that faced them, there is no problem with being ‘switched on’” (p. 11).

There is empirical evidence that affective rumination, problem-solving pondering, and detachment are distinct factors [8]. Moreover, affective rumination and problem-solving pondering are differentially related to different aspects of employee health, well-being and performance. In a cross-sectional study, Cropley and colleagues [8] found that while affective rumination was positively related to choosing unhealthy foods, problem-solving pondering did not correlate with eating behavior. In this study, detachment correlated negatively with choosing unhealthy foods. Furthermore, in a two-wave survey study, Hamesch et al. [20] found that while affective rumination predicted depressive symptoms six months later, problem-solving pondering did not yield detrimental effects. Syrek and colleagues [21] found in a diary study that while affective rumination was positively related to sleep impairment, problem-solving pondering tended to be beneficial to sleep quality. In a longitudinal panel study, Vahle-Hinz et al. [22] provided evidence that while affective rumination predicted lower levels of recovery and did not affect creativity six months later, problem-solving pondering predicted higher levels of creativity but did not impair recovery status. Applying latent profile analysis, Kinnunen and colleagues [23,24] found that different configurations of affective rumination, problem-solving pondering, and (lack of) detachment differentially predicted changes in exhaustion, work engagement, and sleep problems. Finally, another longitudinal study by Firoozabadi and colleagues [25] applied growth curve modeling and found that while affective rumination predicted increases in exhaustion over twelve months, problem-solving pondering was only related to the initial level of exhaustion.

While the studies cited above have consistently shown the differential effects of affective rumination vs. problem-solving pondering, very few studies have investigated affective rumination and problem-solving pondering in concert with psychological detachment for exceptions see [8,23,24,26]. In this sense, although much of the above cited research explicitly refers to detachment, we run the risk of disconnecting research on affective rumination and problem-solving pondering from the large body of research on detachment. At present, it is not possible to draw a clear conclusion regarding the relative roles of each facet of thinking about work for explaining employee well-being. Given correlations of r > |0.50| between detachment and the other facets in the initial studies of the tripartite conceptualization [8,23,24,26], studying the three facets in an integrated way appears imperative to scrutinize the unique contribution of each facet for explaining variance in employee well-being. Research which takes this direction contributes to reunifying research on detachment and the other two facets of rumination empirically.

### 1.3. Beyond the Tripartite Model of Work-Related Rumination—Positive and Negative Work Reflection

The tripartite conceptualization and empirical research on affective rumination and problem-solving pondering have contributed to the literature by pointing out that there are aspects of thinking about work during off-job time which may not be detrimental to employee well-being at all or may even be conducive to well-being. Taking this idea one step further, scholars have introduced the concept of positive work reflection [10]. Positive work reflection refers to thinking about the positive aspects of one’s job. Within Martin and Tesser’s taxonomy of rumination in general positive work reflection may represent or may contain elements of reminiscence [14]. Interestingly, from a conceptual point of view, there are arguments that positive work reflection and problem-solving pondering may have considerable overlap. More specifically, Carver [27] suggests that “reminiscence might represent a problem-solving response to a current dissatisfaction” (p. 58). Put another way, an individual may think about the good sides of the job (e.g., past successes) to regulate negative affect while engaging in problem-solving pondering. On a related note, problem-solving pondering is not confined to trying to find solutions to work-related problems in off-job time, but also refers to actually finding solutions to work-related problems in off-job time [9]. If employees make significant progress through their cognitive problem-solving efforts and experience success, problem-solving pondering may be a source or a trigger of positive work reflection. In this sense, it is likely that the two facets are tightly linked. Thus, it not trivial that positive work reflection is distinct from problem-solving pondering. Accordingly, we will scrutinize the distinction particularly between these two facets of work-related rumination in our focal analyses.

Positive work reflection has been theorized in terms of an opportunity for reappraisal of stressful work situations in off-job time [10]. Accordingly, whereas affective rumination and lack of psychological detachment have been conceptualized as resource-draining experiences, positive work reflection has been theorized as an activity which may contribute to replenishing resources [10]. In line with this perspective, positive work reflection during the weekends or vacations has been shown to predict lower levels of disengagement [10,28], exhaustion, and higher levels pursuit of learning (i.e., seeking opportunities for learning) during the next workweek [10]. Drawing on a series of diary studies, Meier and colleagues [29] provided evidence that positive work reflection is related to affective well-being as reflected in higher levels of joviality, and lower levels of depressive and angry mood. Importantly, they provided evidence that positive work reflection explains incremental variance in affective well-being beyond psychological detachment and absence of negative reflection. Although not perfectly consistent across the three studies, there is evidence that positive work reflection is positively linked to affective well-being both at the within and the between-person level of analysis. Positive work reflection during the weekend also predicted facets of contextual performance, such as personal initiative, creativity, and organizational citizenship behavior during the workweek [11]. Furthermore, a study by Daniel and Sonnentag [30] suggests that positive work reflection is positively linked to work engagement and work-family enrichment.

Whereas positive work reflection appears obviously distinct from detachment, affective rumination, and to some extent also from problem-solving pondering as discussed above, negative work reflection seems to have high conceptual overlap particularly with affective rumination. Researchers from both approaches have drawn on the same theoretical arguments from general rumination research [31] when discussing why either negative work reflection or affective rumination impairs well-being or performance [11,32,33,34]. In essence, negative work reflection refers to thinking about the negative sides of one’s job and realizing what one does not like about it. Conceptually, we see two basic differences between affective rumination and negative work reflection. First, negative work reflection implies taking a more general perspective on job-related events than affective rumination. Second, while affective rumination is focused on highly activated negative affect triggered by work-related thoughts, negative work reflection is focused on cognition rather than affect. It refers merely to instances of thinking about (negative aspects of) work rather than affective reactions to these thoughts. This cognitive focus is in line with cognitive theories of rumination [14], for an in-depth discussion see [35] Negative work reflection during vacation has been linked to higher levels of health complaints, emotional exhaustion, and disengagement, as well as lower levels of effort expenditure after vacation [28]. Interestingly, in the vacation study, effects of positive work reflection did not show a reverse pattern to negative work reflection, but yielded negligible effects on these outcomes [28]. In a similar vein, Binnewies and colleagues [11] found that while positive work reflection predicted contextual performance outcomes, negative work reflection did not. These patterns of results suggest that positive work reflection and negative work reflection are not opposite ends of the same continuum see [12]. Negative work reflection has been shown to be distinct from positive work reflection [11,28,29] and psychological detachment [36]. However, to the best of our knowledge there has been no empirical research comparing negative work reflection to affective rumination and problem-solving pondering. Hence, our study contributes to bridging literatures on the tripartite conceptualization of work-related rumination and work reflection. Drawing on the literature reviewed above, we expect the five facets of work-related rumination to be distinct when applying confirmatory factor analysis. Accordingly, we state:

**Hypothesis** **1.**
*Psychological detachment, affective rumination, problem-solving pondering, positive work reflection, and negative work reflection will emerge as correlated, yet distinct, factors.*


### 1.4. Linking Rumination to Different Facets of Employee Well-Being

Above we have given a brief review of the empirical evidence on the five facets of work-related rumination. Rather than deriving specific hypotheses on differential effects of the five facets, in this section we basically justify why the set of focal outcome variables or correlates is reasonable and which overall pattern of correlations should be expected. In this study, we consider how the five facets of thinking about work relate to burnout, work engagement, thriving, satisfaction with life, and flourishing. While burnout, work engagement, and thriving refer to aspects of work-related well-being, satisfaction with life and flourishing can be considered aspects of general context-free well-being. Satisfaction with life taps into hedonic well-being (or happiness). Flourishing captures eudemonic well-being (or purpose in life).

First, we consider differential relations of the five facets of thinking about work to the emotional exhaustion facet of burnout [37]. Emotional exhaustion refers to low levels of energy and feeling spent [38]. Theoretical approaches to rumination in general and work-related rumination in more specific terms suggest that perseverative cognition is energy-consuming in its own right and will hamper processes of recovery [1,4,11,39]. Meta-analytic evidence suggests that psychological detachment will yield a negative association with exhaustion [6]. For instance, a recent study found a moderate negative association between detachment and exhaustion [40], see also the work by the authors of [41]. Furthermore, from the literature reviewed above, we expect that particularly affective rumination will yield a strong positive association to exhaustion due to its affective focus [23,24,25]. In a recent study Vandevala and colleagues compared the predictive validity of detachment, affective rumination, and problem-solving pondering regarding burnout. They found that neither detachment nor problem-solving pondering explained incremental variance in emotional exhaustion beyond affective rumination [42]. The same should be true for negative work reflection [32]. By contrast, problem-solving pondering should be largely unrelated to exhaustion [23,24,25] and positive work reflection can be expected to yield a negative association [10,28]. Given that psychological detachment is rather neutral in affective terms [6], relations will be weaker than for affective rumination and negative reflection [23,24].

Second, we examine relations between the five facets of work-related rumination and work engagement. Work engagement has been described in terms of “a positive, fulfilling, work-related state of mind that is characterized by vigor, dedication, and absorption” [43] (p. 74). There are non-significant links between psychological and work motivation in the recent meta-analysis [6]. Furthermore, a recent study on work-related well-being tapping into aspects of work engagement [44] found only a very modest link between detachment and work well-being. Hence, we expect no association between detachment and work engagement see also [23]. In our study, we differentiate between the three facets of work engagement and provide more precise insights regarding the detachment-work engagement link. In a similar vein we suggest that affective rumination will at best yield weak to moderate links to work engagement [23]. According to prior empirical evidence [23], problem-solving pondering should be largely unrelated to work engagement. By contrast, the empirical evidence presented above suggests that positive work reflection should be positively related to work engagement [30], as positive work reflection implies high involvement in work [45]. The empirical evidence presented above suggests that negative work reflection can be expected to be largely unrelated to work engagement [11,32].

Third, we consider thriving as a correlate of thinking about work. Thriving encompasses the experience of vitality and learning [46]. Thriving has been conceptualized within positive organizational scholarship [47,48]. There is meta-analytic evidence that thriving is negatively related to indicators of perceived stress and explains incremental variance in subjective health, burnout, and job satisfaction beyond positive affect and work engagement [49]. Psychological detachment, affective rumination, and negative work reflection should be largely unrelated to learning. By contrast, given the links between positive work reflection and pursuit of learning cited above [10], we expect that particularly positive work reflection is linked to the learning facet of thriving. A similar rationale might apply to problem-solving pondering. The vitality facet of thriving implies high levels of human energy [50,51]. Empirical evidence suggests that psychological detachment [29] may tend to relate positively to vitality. Given the energy-draining nature of affective rumination, [25] it should be negatively related to the vitality facet of thriving. Problem-solving pondering implies that employees come up with new ideas or find solutions to work-related problems in off-job time [9]. Hence, energizing effects regarding vitality are highly plausible from a theoretical perspective [52]. Following the idea that positive work reflection is associated with gaining resources [10] and in line with prior empirical evidence on joviality [29], we expect a positive link between positive work reflection and vitality. Empirical evidence on links between negative work reflection on the one hand and vigor [32] and joviality [29] on the other hand is inconsistent and suggests that negative work reflection will be largely unrelated to vitality.

Fourth, we consider relations to satisfaction with life [53]. Meta-analytic evidence suggests that detachment will be positively related to satisfaction with life [6]. Given that the links between the other forms of work-related rumination and satisfaction with life have not been studied empirically, we speculate that affective rumination—due to its negative affective quality—will yield a negative association with satisfaction with life. Because problem-solving pondering may also include making progress towards significant personal goals [52,54], we suggest that problem-solving pondering may be positively associated with satisfaction with life. Given that research on self-reflection in general suggests no links to satisfaction with life, positive and negative work-reflection might eventually yield only modest links with satisfaction with life.

Fifth, flourishing or psychological well-being has been conceptualized as the eudemonic component of well-being [55]. Eudemonic well-being is rooted in humanistic theories of positive functioning and encompasses experiencing purpose in life, meaning, self-acceptance, and a sense of competence [55,56]. Although eudemonic well-being has been conceptualized as a multifaceted phenomenon [56], we follow Diener and colleagues’ approach [55] and study flourishing as an overall construct which combines different aspects of eudemonic well-being from different conceptualizations into an overall score. Given that, to the best of our knowledge, there is no empirical research on work-related rumination and flourishing, we explore the links between the five facets of rumination and flourishing in our study. We speculate that particularly positive work reflection, as an aspect of self-reflection, might contribute to a sense of self-acceptance and competence and, hence, yield positive associations with flourishing, see the works by the authors of [57,58,59] for evidence on related concepts.

In sum, based on the line of reasoning presented above, we expect the links of the five facets of work-related rumination to yield differential associations with the focal well-being outcomes. Although we have elaborated on the specific patterns we expect to find above, we state a general hypothesis on these associations as follows:

**Hypothesis** **2.**
*Psychological detachment, affective rumination, problem-solving pondering, positive work reflection, and negative work reflection relate differentially to emotional exhaustion, work engagement, thriving, satisfaction with life, and flourishing.*


### 1.5. Concomitants of Thinking about Work

To further clarify the nomological network surrounding work-related rumination and related concepts in this study, we consider three additional variables. These variables have been considered in the context of work-related rumination and may overlap to some extent with the focal constructs. Given that our focus is on the five facets of work-related rumination discussed above and how they differentially relate to employee well-being, we keep discussion of additional correlates brief and provide the results in terms of supplemental rather than focal analyses.

First, the concept of irritation seems to have considerable overlap with some of the five facets of thinking about work in leisure time. Irritation has been conceptualized as a strain reaction or a state of mental impairment closely linked to the job [60]. Two facets of irritation have been distinguished, namely cognitive and emotional irritation. The first facet of irritation, cognitive irritation, has been labeled and defined in terms of rumination in some publications [60,61,62]. Items measuring cognitive irritation refer to trouble switching off after work and recurrent thoughts about work while at home or during vacations. Mapping cognitive irritation on the tripartite conceptualization discussed above [8,9] suggests that it corresponds more closely to low levels of psychological detachment, rather than affective rumination. The second facet of irritation, emotional irritation, has been conceptualized in terms of irritability. Focal items measuring emotional irritation basically tap into tense arousal as a reaction to goal discrepancies [60]. Given that emotional irritation concurs with affective rumination in focusing on highly activated negative affect (e.g., feeling nervous or irritated), we expect that emotional irritation should correspond closely to affective rumination in the tripartite conceptualization. Given the considerable overlap of the two facets of irritation particularly with lack of detachment and affective rumination, considering links between irritation and the five facets of thinking about work will facilitate integration of research on irritation and the tripartite conceptualization of work-related rumination [61].

Second, we examine associations of the five facets of work-related rumination with worry. In general terms, worry refers to thoughts about future negative events, which are hard to control or dismiss [63]. In our view, the unique aspect of worry vs. rumination is that it is future-oriented. However, in clinical psychology “rumination and worry are theorized to share key qualities of unpleasantness, repetitiveness, abstractness, and uncontrollability of thinking” [64] (p. 926). The two concepts have been studied extensively mostly in clinical, social, and personality psychology, and some studies have compared worry and rumination. There is evidence that worry and rumination in general (nonwork-related) terms are highly correlated (0.45 < r < 0.62) yet distinct constructs e.g., [65,66,67,68,69,70,71]. Interestingly, research on facets of work-related rumination has not explicitly distinguished between rumination and worry. Whereas work reflection is focused on past events, psychological detachment, affective rumination, and problem-solving pondering are unspecific about the timing of triggers (past, present, future). To facilitate integration of work-related rumination with a clinical psychology perspective, we investigate the strength of associations between the five facets of work-related rumination and worry in general terms.

Third, we consider neuroticism as an aspect of personality that can be expected to be linked to the facets of work-related rumination [35]. Neuroticism is one of the widely studied big five personality traits [72]. Individuals high on neuroticism (or low on emotional stability) tend to be, for instance, easily irritable, moody, nervous, self-pitying, anxious, and fretful [73]. Hence, neuroticism may predispose individuals to ruminate in general terms and more specifically about work-related issues. In line with this line of reasoning, meta-analytic evidence suggests that neuroticism is moderately negatively related to psychological detachment [6]. Hamesch and colleagues [20] found that neuroticism yields a medium-sized correlation with affective rumination and moderate associations with problem-solving pondering. In a diary study, Wiese and colleagues [74] found a medium-sized correlation between (trait) neuroticism and daily rumination about difficult client interactions at the between-person level (Their two-item measure corresponds most closely to lack of psychological detachment). By contrast, however, Wang and colleagues [35] found no significant association between neuroticism and rumination about customer mistreatment at the between-person level. (Their measure tapped into aspects of lack of detachment and affective rumination due to customer mistreatment.) In sum, these findings suggest that the strength of association between neuroticism is dependent upon the facet of work-related rumination considered. Hence, studying differential links between neuroticism and the five facets of work-related rumination provides a more detailed picture of the patterns of associations and has the potential to qualify or even reconcile inconsistent evidence from prior research.

To the best of our knowledge, there is almost no empirical evidence combining facets of work-related rumination and its concomitants. Although Ebert and colleagues [75] provide evidence that the beneficial effects of a recovery intervention differentially change levels of detachment, irritation, and worry (effect sizes of the intervention varies from variable to variable), they do not provide correlations among the variables and they do not discuss whether the variables are distinct empirically. In sum, there are good reasons to expect the five facets of work-related rumination to be related to irritation, worry, and neuroticism. Importantly, however, our review of the literature suggests that the five aspects of work-related rumination will be distinct from the five facets of work-related rumination. We state:

**Hypothesis** **3.**
*Psychological detachment, affective rumination, problem-solving pondering, positive work reflection, and negative work reflection will emerge as factors distinct from irritation, worry, and neuroticism.*


## 2. Materials and Methods

### 2.1. Procedure

We conducted a cross-sectional survey study online. The survey was announced as a study on job demands and work-related thoughts. The procedure and materials of this study have not undergone examination by an ethics committee, as the measures and procedures of our study followed the protocols of standard self-report survey research in applied psychology, and we did not touch sensitive topics (e.g., sexual orientation). Our study fully complied with the standards of the Department of Psychology at the University of Hagen, which included strict guidelines to guarantee anonymity of the self-reported data. Individuals interested in participating in our study were informed about the general aims and the protocol of the study before their participation. Our protocol did not include any form of deception of participants. Participation was voluntary and participants had the opportunity to quit whenever they wanted.

### 2.2. Sample

Our sample consisted of employees enrolled in a psychology program at a German university that offers distance-learning courses. Our initial dataset consisted of 508 participants. After eliminating cases because of missing answers, lack of seriousness or unrealistic low processing time, the final dataset included 474 participants. The full sample consisted of 72 per cent women and 28 per cent men. The average age in the focal sample was 34.34 years (*SD* = 9.55), ranging from 19 to 72. The major proportion of the participants had either general qualifications for university entrance (84%) or advanced technical college entrance qualifications (8%). On average, participants worked 29 h per week (*SD* = 11.3). Four hundred twenty persons worked within a team and the majority of participants did not have a managerial position (77%). The majority (399 persons) had a permanent employment contract.

### 2.3. Measures

We applied a consistent response format for all facets of thinking about work. In line with the original work on the tripartite rumination scale [8] we applied a five-point Likert-type scale (1 = very seldom or never; 2 = seldom; 3 = sometimes; 4 = often; 5 = very often or always) and applied this format to the work reflection items as well. We followed this strategy to rule out the possibility that distinct factors might emerge due to methodological (rather than conceptual) differences between the focal scales.

#### 2.3.1. Psychological Detachment

We measured psychological detachment, affective rumination, and problem-solving pondering using the validated 15-item-questionnaire Work-Related Rumination Scale (WRRS) from Cropley, Michalianou, Pravettoni, and Millward [8], which has been adapted to German [21]. We measured psychological detachment with five items. A sample item is “Do you leave work issues behind when you leave work?”

#### 2.3.2. Affective Rumination

We applied five items of the WRRS to measure affective rumination [8,21]. A sample item is “Do you become tense when you think about work-related issues during your free time?”

#### 2.3.3. Problem-Solving Pondering

We applied five items of the WRRS to measure problem-solving pondering [8,21]. A sample item is “I find solutions to work-related problems in my free time.”

#### 2.3.4. Positive Work Reflection

We measured positive work reflection using the 3-item-scale developed by Fritz and Sonnentag [10]. The items referred to how often positive thoughts about work occur during leisure. A sample item is “During leisure time, I consider the positive aspects of my job.”

#### 2.3.5. Negative Work Reflection

We measured negative work reflection using the 3-item-scale developed by Fritz and Sonnentag [28] The items referred to how often negative thoughts about the job arise after work. A sample item is “During leisure time, I consider the negative aspects of my job.”

#### 2.3.6. Burnout

Participants rated their experienced degree of physical and psychological fatigue and exhaustion using the subscale personal burnout of the Copenhagen Burnout Inventory [76], which consists of eight items. Responses ranged from 1 (less than once a month) to 5 (several times a day). We asked participants to refer to the period of the last three months. Sample items are “How often are you physically exhausted?” or “How often do you feel worn out?”

#### 2.3.7. Work Engagement

We applied nine items from the Utrecht Work Engagement Scale (UWES-9) developed by Schaufeli, Bakker and Salanova [77] to capture work engagement. The response format ranged from 1 (never) to 7 (always). The UWES-9 is divided into three subscales with three items each. The scale has been adapted to German [78]. Vigor measures energy and pleasure at work. A sample item is “At my job, I feel strong and vigorous.” The second facet dedication sums up the extent of pride, enthusiasm and inspiration in the form. A sample item is “I am enthusiastic about my job.” The third subscale absorption points to immersion into work. A sample item is “When I am working, I forget everything else around me.” Responses ranged from 1 (never) to 7 (always).

#### 2.3.8. Thriving

Thriving has been conceptualized as the shared experience of vitality and learning at work. We measured thriving with ten items from Porath, Spreitzer, Gibson, and Garnett [46]. The items have been adapted to German [79]. Five items referred to vitality and five items referred to learning. Response format ranged from 1 (strongly disagree) to 7 (strongly agree). A sample item for vitality is “At work, I feel alive and vital.” A sample item for learning is “At work, I see myself continually improving.”

#### 2.3.9. Satisfaction with Life

We applied a validated German adaption of the Satisfaction with Life scale [80] to measure satisfaction with life [81]. Response format ranged from 1 (strongly disagree) to 7 (strongly agree). A sample item is “In most ways my life is close to my ideal.”

#### 2.3.10. Flourishing

We applied the respective items from the New Well-Being Measures [55] to measure flourishing. The scale has been adapted to German [82]. We applied the full scale consisting of eight items. Response format ranged from 1 (strongly disagree) to 7 (strongly agree). A sample item is “I lead a purposeful and meaningful life.”

In addition to the focal aspects of thinking about work in leisure time and the focal indicators of well-being we included three constructs regularly discussed when referring to (work-related) rumination: worry, irritation, and neuroticism.

#### 2.3.11. Irritation

We applied the eight items of the Irritation Scale [83] to capture irritation. The irritation scale has been developed in German. Several adaptions [60] for different languages exist. Three items capture the cognitive facet of irritation (“rumination”) connected to problems detaching from the job. A sample item is. “Even at home I often think of my problems at work.” Five items captured the emotional facet of irritation (“irritability”), which basically taps into experiences of tense arousal due to work. A sample item is “I get grumpy when others approach me.” Responses ranged from 1 (never/rarely) to 5 (very often/always).

#### 2.3.12. Worry

We applied the German adaption [84] of the Penn State Worry Questionnaire [85] to measure worry. The scale contains 16 items to measure the tendency of having worried thoughts. A sample item is “I worry about projects until they are all done.” Responses ranged from 1 (never/rarely) to 5 (very often/always).

#### 2.3.13. Neuroticism

We measured neuroticism applying items from the Mini-Markers by Saucier [73]. The German version was validated by Weller and Matiaske [86]. The subscale for neuroticism consists of eight adjectives (e.g., “jealous”, “touchy”, or “moody”). Responses ranged from 1 (strongly disagree) to 7 (strongly agree).

### 2.4. Analytic Strategy

Given the considerable sample size, we analyzed the data leveraging structural equation modeling [87]. We employed the “lavaan”-package in R [88]. Throughout all models, we applied diagonally-weighted least squares (rather than maximum likelihood) estimation to avoid problems associated with violation of the assumption of multivariate normal distribution of data (e.g., skewness or kurtosis of scales) [89]. We report standardized estimates throughout all analyses.

Statistical analysis was done in two steps. First, we examined the psychometric properties of the focal scales. We focused on confirmatory factor analyses comparing competing models. In essence, this part of the analysis refers to the measurement models in structural equation models. Second, we added regression paths from each facet of work-related rumination to each aspect of employee well-being. This part of the analysis adds the structural model. Notably, the second step of specifying a (latent) structural equation model is superior to analyzing bivariate correlations or covariance between the focal variables. Our full structural equation model allows examining the relative roles of each facet of work-related rumination when entered simultaneously with the other facets while predicting the full set of focal outcomes. Our analyses, hence, allow for quantifying to what extent one facet of work-related rumination (e.g., positive work reflection) contributes to accounting for variance in satisfaction with life beyond the other facets of work-related rumination.

## 3. Results

### 3.1. Factorial Structure and Psychometric Properties of the Focal Scales

The means, standard deviations, alpha reliabilities, and zero-order correlations between the focal variables are presented in Table 1. Beyond this descriptive perspective the first aim was to examine whether the five facets of work-related rumination can be considered as distinct rather than redundant aspects of thinking about work. Furthermore, we wanted to gain insights into the degree of association between the different facets. Hence, we specified different confirmatory factor analysis models starting with a 1-factor model, in which all items loaded on one common factor. In the next step, we specified our focal five factor model, consisting of psychological detachment, affective rumination, problem-solving pondering, positive work reflection, and negative work reflection. As a standard of comparison, we also specified more parsimonious two-, three-, or four-factor models in which items from different scales loaded on a common factor. Rather than applying arbitrary combinations, we combined facets which might overlap considerably from a conceptual point of view. The focal models are illustrated in Figure 1. We ran several confirmatory factor analysis models and compared the focal 5-factor model with the alternative models applying a set of indicators of model fit. In the 5-factor model on the left items of work-related rumination were specified to load on five distinct factors in the way these items have been conceptualized. In the alternative models in the middle, we combined items from different facets to load on a common factor. For instance, in the 4-factor model a, we combined items of affective rumination and items of negative work reflection to form a common factor. Finally, we examined the single factor model presented on the right, in which all items of work-related rumination loaded on one factor. The results of these comparisons are presented in Table 2. We followed recommendations regarding fit-indices and corresponding cut-off criteria by Schermelleh-Engel, Moosbrugger, and Müller [90]. Recommended cut-off criteria are summarized in Table 3. In essence, we found that the five-factor model fitted the data better than any of the alternative confirmatory factor analysis models tested (Delta *χ*^2^ ≥ 167.97). The five-factor model has an acceptable fit to the data, as evidenced in CFI = 0.961, TLI = 0.954, RMSEA = 0.071, and SRMR = 0.082. In sum, our analyses provide evidence that distinguishing between the five facets of psychological detachment, affective rumination, problem-solving pondering, positive work reflection, and negative work reflection is warranted.

### 3.2. Differential Links between the Facets of Rumination and Employee Well-Being

To provide a clearer picture of how the five focal facets of work-related rumination relate differentially to the set of well-being variables, we specified structural equation models. Beyond the confirmatory factor analysis presented above (our measurement model), we added regression paths from each facet of work-related rumination to each facet of employee well-being. The path coefficients reflect the standardized covariance among the five facets of work-related rumination on the left and the standardized regression coefficients for each combination of the five facets of work-related rumination and the eight indicators of employee well-being. Standardized coefficients may range from −1.0 to 1.0. Hence, it is easy to infer the strength of association among the latent factors. Applying latent variables structural equation modeling accounts for measurement error and hence reflects the estimated true association among variables. Our focal model yielded acceptable to good fit as evidenced in CFI = 0.982, TLI = 0.980, RMSEA = 0.041, and SRMR = 0.064. The results are presented in Figure 2. For ease of comprehension, we display coefficients for each combination of predictors and outcomes. Importantly results refer to coefficients from the complete structural equation model including all predictors and outcome variables at the same time.

A closer inspection of the standardized covariances among the five facets of work-related rumination presented on the left side of Figure 2, suggests that the five constructs share considerable portions of variance. However, they emerge as distinct factors and yield differential links to the set of outcomes considered. We report the results below for each of the five facets of work-related rumination.

Psychological detachment explains unique variance in almost all the well-being facets considered. Psychological detachment is the strongest predictor for the general well-being indicators of flourishing (β = 0.60) and satisfaction with life (β = 0.40). Psychological detachment also contributes to predicting the work engagement facet of vigor and the two facets of thriving (learning β = 0.26; vitality β = 0.32). By contrast, detachment explains only small portions of incremental variance in burnout (β = −0.11) and does not predict dedication and absorption beyond the other facets of work-related rumination. This finding is important, given that psychological detachment yields bivariate correlations with almost all outcomes.

Affective rumination explains unique variance particularly in burnout (β = 0.38) and also predicts vigor (β = −0.20), dedication (β = −0.15), and the vitality facet of thriving (β = −0.16). However, affective rumination does not contribute to explaining additional variance in the work engagement facet of absorption, the learning facet of thriving, satisfaction with life, and flourishing beyond the other facets of work-related rumination. These results qualify the bivariate relationships with all outcomes of work-related well-being.

Problem-solving pondering contributes particularly strongly to predict the work engagement facet of vigor (β = 0.40), the learning facet (β = 0.49), and the vitality facet of thriving (β = 0.44). Problem-solving pondering also explains incremental variance in the other facets of work engagement (dedication β = 0.34; absorption β = 0.33) and the indicator of general well-being, namely, satisfaction with life (β = 0.32) and flourishing (β = 0.34). Hence, problem-solving pondering is predictive of all facets of employee well-being considered in this study, except for burnout.

Positive work reflection does not contribute to predicting burnout. However, positive work reflection emerges as a unique predictor of all the other (positively connotated) well-being outcomes in our study. Associations are particularly strong for the work engagement facets (vigor β = 0.38; dedication β = 0.49; absorption β = 0.49) and the vitality facet of thriving (β = 0.37). Importantly, positive work reflection explains unique variance in the learning facet of thriving (β = 0.24), satisfaction with life (β = 0.16), and flourishing (β = 0.20) as well.

Negative work reflection explains unique variance in all well-being outcomes considered, except for flourishing. The largest coefficients emerged for predicting the facets of thriving (learning β = −0.35; vitality β = −0.35) and work engagement (vigor β = −0.31; dedication β = −0.31; absorption β = −0.29). Negative work reflection also explains variance in burnout (β = 0.22) and satisfaction with life (β = −0.19).

### 3.3. Associations between the Five Facets of Work-Related Rumination and Their Concomitants

In the next step, we examined the extent to which the five facets of work-related rumination can be distinguished empirically to similar constructs. We ran a set of confirmatory factor analyses to scrutinize whether items of the five facets of thinking about work vs. irritation, worry, and neuroticism load on distinct factors. Given that irritation consists of two subscales, namely cognitive and emotional irritation, we specified a 9-factor model, including the (1) psychological detachment, (2) affective rumination, (3) problem-solving pondering, (4) positive work reflection, (5) negative work reflection, (6) cognitive irritation, (7) emotional irritation, (8) worry, and (9) neuroticism. As a standard for comparison, we also specified a single factor model, a 3-factor model, a 7-factor model, and two 8-factor models, where items from different scales were combined to form common factors. We followed a similar approach as in our confirmatory factor analyses in Section 3.1. In our focal 9-factor model, all items loaded on their respective factors. We created five alternative models. In the alternative models we combined items from different scales to load on a common factor. For instance, in the 8-factor model a, psychological detachment and cognitive rumination formed a common factor. Figure 3 illustrates the specific alternative models. We compared the models using the chi^2^-square difference test. We also present comparisons of indices of absolute and relative model fit (see Table 3 for cut-off values). We present an overview of the fit indices and the results of the comparison in Table 4. Comparing the fit indices of the competing models suggests that the 9-factor model provides the best fit to the data when compared to the alternative models (delta *χ*^2^ ≥ 64.60). Consistently across absolute and relative model fit, the 9-factor model is superior to the alternative models. The 9-factor model has an acceptable to good fit to the data, as evidenced in CFI = 0.984, TLI = 0.983, RMSEA = 0.040, and SRMR = 0.063. Hence, although cognitive irritation and psychological detachment yield very high bivariate correlations (*r* = −0.77, see Table 1), our results suggest that the two variables can be distinguished empirically. Of note, however, is the fact that the 8-factor model combining psychological detachment and cognitive irritation fit almost equally well according to CFI, TLI, RMSEA, and SRMR.

## 4. Discussion

In this study, we set out to provide insights into the structure of work-related rumination. Our focus was on whether the five major facets of thinking about work in off-job time can be distinguished empirically and how they differentially relate to several facets of work-related and general well-being. Finally, we examined how the five facets of work-related rumination relate to similar constructs conceptualized in clinical and personality psychology.

### 4.1. Theoretical Implications

First, our results aid researchers interested in the structure of work-related rumination. Our findings suggest that although psychological detachment, affective rumination, problem-solving pondering, positive work reflection, and negative work reflection share a considerable portion of variance, they should be treated as distinct constructs. For one, we replicate the evidence on the construct validity of detachment, affective rumination, and problem-solving pondering [8,23,24,26,42]. For the other, we extend empirical evidence beyond the tripartite conceptualization and provide evidence that neither positive nor negative work reflection is redundant in the context of the more established constructs. Whereas positive work reflection correlates only modestly with the other facets of work-related rumination, associations between negative work reflection and the other facets are considerably higher. Yet, considering negative work reflection besides affective rumination or psychological detachment is worthwhile. Our analysis of the overlap specifically between positive work reflection and problem-solving pondering suggests that the two constructs are clearly distinct and correlate only moderately.

Second, our results contribute to extending knowledge on how different types of thinking about work in off-job time relate to a broad range of well-being variables ranging from work-related aspects such as burnout and work engagement to indicators of general well-being, such as satisfaction with life. In other words, we contribute to providing empirical evidence on the convergent and discriminant validity of the five focal facets of work-related rumination. Importantly, we provide an integrative perspective that allows for comparing the relative relevance of each facet of work-related rumination to a broad set of well-being outcomes. Within this broad range of outcome variables, we have included variables that have not been linked to work-related rumination in prior research (e.g., flourishing). Our analysis of associations between the five facets of work-related rumination on the one hand and aspects of employee well-being on the other hand suggests that the predictive power of the five facets varies considerably. Burnout is best predicted by combining affective rumination, negative work reflection, and psychological detachment. These findings qualify and extend the bivariate association between detachment [6,40,41] and affective rumination [23,24,25], and problem-solving pondering [42] with exhaustion found in prior research. Whereas Vandevala et al. [42] found that detachment adds little to predict exhaustion beyond affective rumination, we found that detachment explains at least a modest portion of additional variance in burnout beyond affective rumination. Furthermore, we extend the literature by showing that negative work reflection contributes uniquely to predicting burnout as well, and should be considered in future research to improve prediction of burnout. By contrast, problem-solving pondering and positive work reflection do not contribute to explaining additional variance in burnout—a finding consistent with prior evidence [42]. In this sense, negative aspects of thinking about work (or the lack of detachment from work) seem to be more relevant with regard to burnout than positive ways of thinking about work (i.e., problem-solving pondering and positive work reflection). Consistent with this finding, problem-solving pondering and positive work reflection emerge as the strongest predictors of positive energetic states, such as work engagement and vitality. While almost all facets of work-related rumination contributed to predict thriving, problem-solving pondering yielded the strongest unique predictive power. In this sense, problem-solving pondering is associated with experiencing the self as growing and constantly improving—an aspect of eudemonic well-being [48]. While detachment explains rather modest portions of variance in some facets of energetic well-being (particularly burnout and work engagement), it emerged as the most important predictor of general (nonwork-related) well-being as reflected in satisfaction with life and flourishing. Given that (differential) links between the five facets of work-related rumination and particularly work engagement, thriving, satisfaction with life, and flourishing have rarely been considered in prior research (many combinations have not been studied at all), our study is the first to provide a comprehensive picture of the strength of association between work-related rumination and employee well-being.

Third, our results contribute to locating each facet of work-related rumination more precisely within the nomological network surrounding perseverative thought and related concepts, such as irritation, worry, and neuroticism. We provide evidence that all facets of work-related rumination are empirically distinct from their concomitants from other streams of research. These results confirm that the five facets or work-related rumination capture aspects distinct from dispositions and traits studied in clinical and personality psychology. In line with our expectations, the strength of association between work-related rumination and the concomitants varied considerably as a function of facet. Some of the facets of work-related rumination were only modestly correlated with worry and neuroticism (particularly positive work reflection). Affective rumination and negative work reflection yielded the strongest links to worry and neuroticism. This finding confirms prior evidence on affective rumination and neuroticism [20]. Of note if the fact that the associations between neuroticism to psychological detachment, affective rumination, and problem-solving pondering are very similar to correlations reported in prior research [20,74]. However, the strength of association varies considerably from facet to facet—a finding that may help explain why some prior studies have failed to find support for significant links to at least some aspects of rumination [35]. Obviously, the strength of association is dependent upon the content and the specific operationalization of work-related rumination. Associations between the cognitive facet of irritation and psychological detachment were very high, too. This finding suggests that the common practice of applying or adapting irritation items to capture (lack of) psychological detachment from work, e.g., the work by the authors of [61] is probably reasonable and warranted, see also similar effect sizes for detachment and irritation in [75]. Moreover, both facets of irritation yielded strong associations with affective rumination. This finding suggests that labeling irritation in the literature as “rumination” [60] may be quite an accurate description as well. In this sense, our study contributes to integrate research on irritation within the research on the five facets of work-related rumination. Our results differ from prior evidence in clinical psychology, where rumination and worry have often yielded very strong links [69]. We believe that the rather modest associations with worry in our study are due to the fact that the five facets refer to work-related thoughts, while worry refers to negative thoughts in more general terms. The same is true for neuroticism.

In sum, our study is the first to provide evidence that the five major facets of work-related rumination, namely psychological detachment, affective rumination, problem-solving pondering, positive work reflection, and negative work reflection are non-redundant concepts. Furthermore, our study contributes to integrating research on the tripartite conceptualization of work-related rumination with research on positive and negative work reflection. We are the first to provide a complete picture of the associations among the five facets of work-related rumination and provide insights into the links between different aspects of work-related rumination and a comprehensive set of facets of employee well-being. Finally, we connect work-related rumination to the nomological networks surrounding concepts from clinical and personality psychology, such as worry and neuroticism. In this sense, our results pave the way for a clearer conceptual map around these constructs and hence help to tailor measures to the purposes of more specific studies on work-related rumination and rumination in more general term.

### 4.2. Practical Implications

Our study is of particular value to scholars conducting empirical research on work-related rumination, because our comprehensive analysis provides evidence-based guidance on the choice of scales tailored to the purposes of a study in the planning phase. Our results suggest that while one specific set of facets of work-related rumination may be well-suited to predicting impaired employee well-being (e.g., burnout), another set of predictors will best predict positive energetic states, such as work engagement and thriving. In this sense, our results suggest that interventions aimed at reducing burnout may be more effective when targeted at reducing negative aspects of work-related rumination (i.e., affective rumination and negative work reflection) rather than positive aspects. By contrast, interventions targeted at improving general well-being in terms of satisfaction with life and flourishing may try to foster switching off rather broadly (i.e., psychological detachment) [91] or encouraging positive ways of thinking about work (i.e., positive work reflection) [29,92].

### 4.3. Strengths and Limitations

Our study features a number of strengths. For instance, we have applied structural equation modeling techniques for our focal analyses to fully scoop the richness of the data from a large and organizationally diverse sample. Furthermore, we have drawn on a set of highly reliable items and scales as evidence in the high alphas presented in Table 1 and consistently high factor loadings. Importantly, we have kept instructions and response formats for all work-related rumination and concomitant scales constant to rule out the possibility that distinct factors emerge purely due to methodological reasons. However, we need to acknowledge a couple of shortcomings that limit the conclusions to be drawn from our results.

First the cross-sectional research design does not allow for conclusions regarding causality. Hence, although, we have discussed our results in terms of work-related rumination affecting employee well-being, there may be reverse (well-being affects levels of rumination) or reciprocal effects at work. However, the proposed direction of causality is in line with theoretical arguments for perseverative thought as a predictor of health impairment [1,93]. Furthermore, we cannot rule out that the associations among the focal variables are inflated or even caused by third variables, such as common method bias [94]. However, in our view, common method variance does not threaten the validity of the results regarding the structure of work-related rumination (confirmatory factor analyses). More specifically, common method bias would render tests of distinct factors more conservative. In other words, it would be harder to find evidence for distinct factors rather than a common factor. The same rationale applies to comparing the relative strength of predictive validity of the five facets of work-related rumination. Furthermore, it has been argued that (common) method bias is unlikely to universally effect a comprehensive set of associations [95]. Time-lagged data or longitudinal data would be preferable for examining causality. However, panel studies or intensified research always requires confining to a limited set of scales and items [96] and often comes at the cost of (considerably) small(er) sample sizes. Moreover, even cross-lagged panel studies do not provide for unambiguous evidence regarding causality, and, strictly speaking, only experimental designs allow for firm conclusions [97]. The correlations presented in this study provide insights into the probable range of effect sizes and help in planning studies applying optimal lags in the future [98]. Intensified longitudinal designs are also better suited to investigating how long it takes for the effects of work-related rumination on well-being to unfold over time applying continuous time structural equation modeling [99]. Our study contributes to better conceptual clarity as a first step.

Second, although we have included a comprehensive set of variables in our analysis, there might be other operationalizations (e.g., detachment applying the recovery experience questionnaire [5]), other concomitants (e.g., mind-wandering, i.e., thinking about off-job topics while at work [100]), or an extended set of outcome variables (e.g., depression) worthwhile to consider in concert with the five facets our study was focused on. As in every empirical study, we had to narrow the scope to a manageable set of variables. We believe that our focus – albeit limited – is reasonable in light of prior research, and provides an important step forward.

Third, our study is limited to self-reported data from a single source. However, our focus was the examination of self-report measures of work-related rumination from the perspective of the individual. Furthermore, peer reports of internal states, for instance by peers or significant others, may not be as valid as self-reports. At the same time capturing multisource data requires considerable additional effort and caution upon collecting data (e.g., applying snowball sampling [101]). However, ours is meant to be first step to explore patterns of associations among a comprehensive set of variables. In this case, cross-sectional survey designs are recommended in the organizational research methods literature [97]. Furthermore, we followed steps recommended in the literature [94] for minimizing method bias by survey design, such as applying different response formats for work-related rumination vs. well-being, separating these variables within the survey, emphasizing anonymity of response, and avoiding ambiguity in our materials.

Fourth, our sample consists of a convenience sample. Although, this sample is diverse in terms of branches, organizations, and occupations, we cannot be sure that our results will generalize to even broader contexts. Our sample is confined to highly educated individuals from Germany. Hence, we cannot be sure that the results will generalize to employees from less privileged contexts (e.g., clickworkers) and other cultures (e.g., Asia). However, we have applied adapted and validated scales throughout the study, and we found results consistent with findings from prior research, conducted particularly in Europe e.g., [8] and the U.S. [13].

In sum, we believe the study—albeit imperfect with regard to many aspects—offers valid, meaningful, and important results concerning the focal questions on the structure or work-related rumination.

### 4.4. Avenues for Future Research

While our results provide new and interesting insights into the structure of work-related rumination, a couple of new questions emerge from our analysis.

We focused our analysis of the structure of work-related rumination on chronic or trait levels of work-related thoughts in off-job time. However, much of the prior research on the facets of work-related rumination has applied experience sampling methodology, analyzing intraindividual associations between variables over time [96,102]. Hence, while our results refer to the between-person level of analysis, research on the structure of work-related rumination may not necessarily come to the same conclusions at the within-person level of analysis. Although, there is empirical evidence that correlations tend to be homologous across levels for many variables studied in occupational stress research [103], future research should scrutinize the differential effects of different facets of work-related rumination at the within-person level, too.

On a related note, in the above, we have acknowledged that our cross-sectional study precludes inference of causal links between work-related rumination and employee well-being. Future research may build on the insights gained from our study to investigate cross-lagged unidirectional or reciprocal links between work-related rumination and employee well-being over time [24]. Importantly, the five facets of work-related rumination may not merely differ with regard to the strength of effects, but also with regard to the time it takes until causes actually take effect [15,16,98]. Our results provide insights into effect sizes and help to determine optimal lags in time-lagged studies of predictive validity or longitudinal research.

In the above study, we have outlined that the associations between the five facets of work-related rumination and employee well-being may be inflated or even be caused by third variables. Given that many countries, for instance in Europe, currently face economically difficult times or are threatened by an economic recession, taking the economic context into consideration when studying work-related rumination may be worthwhile. More specifically, following the 2008 financial crisis, researchers [104] have examined links between the onset of economic crisis and several indicators of mental health. For instance, Giorgi and colleagues [105] found that fear of the crisis yields associations with mental health in terms of loss of confidence, social dysfunction, anxiety, and depression. Future research may consider whether experiencing times of economic crisis covaries with the five facets of work-related rumination. More importantly, identifying differential patterns of association would be interesting for connecting the five facets to the sociological and economics literature. We speculate that aspects of work-related rumination like affective rumination might be more sensitive to the economic context than rumination in more general terms.

Given that the five facets of work-related rumination have been derived from very different lines of theorizing and have been studied mostly in isolation (e.g., studies on detachment only), future research should aim for a better integration of these aspects in conceptual terms. Our study provides a first step for gaining a clearer picture of how detachment, affective rumination, problem-solving pondering, positive work reflection, and negative work reflection are related to one another. However, some of the focal scales have considerable overlap and do not provide for non-redundant coverage of work-related rumination. On a related note, most items of work-related rumination considered here combine different aspects within the same scale that one might prefer to measure separately. For instance, over 20 years ago Martin and Tesser [14] provided a taxonomy of ruminative thought, distinguishing forms of rumination with regard to valence of thoughts and motivational orientation (attainment vs. discrepancy). They also made a case for distinguishing different types of ruminative thought regarding time perspective (past, present, and future). Accordingly, time perspective is reflected in the distinction between worry and rumination in clinical, social, and personality psychology [63]. In this sense, worry is focused on future-oriented thoughts, and rumination refers to experiences from the past. Unfortunately, items of detachment, affective rumination, problem-solving pondering, and, to some extent, also work reflection, are unspecific about whether thoughts are focused on retrospection of past experiences or anticipation of the future. Although this distinction may not make a big difference from a practical point of view (employees refrain from switching off in any case), being more precise in measuring work-related rumination may contribute to gaining better insights into what people actually have on their minds, and why, when their mind is working overtime.

## 5. Conclusions

In our study, we set out to analyze the structure of work-related rumination drawing on the five major facets and scales applied in the literature on applying structural equation modeling to cross-sectional survey data. First, our analyses provide evidence that (1) psychological detachment, (2) affective rumination, (3) problem-solving pondering, (4) positive work reflection, and (5) negative work reflection form five empirically distinct factors. Second, we show that the five facets of work-related rumination relate differentially to burnout, work engagement, thriving, satisfaction with life, and flourishing. While affective rumination emerges as the strongest unique predictor of burnout, problem-solving pondering was the strongest predictor of positive energetic well-being. Psychological detachment was a particularly strong predictor of satisfaction with life and flourishing. Third, we present evidence that the five facets of work-related rumination are distinct from irritation, worry, and neuroticism, although detachment yielded particularly a high link to cognitive irritation. In sum, our study provides a modest, yet important, first step towards the integration of the diverse facets of thinking about work in off-job time within a coherent taxonomy of work-related rumination.

## Figures and Tables

**Figure 1 ijerph-16-02987-f001:**
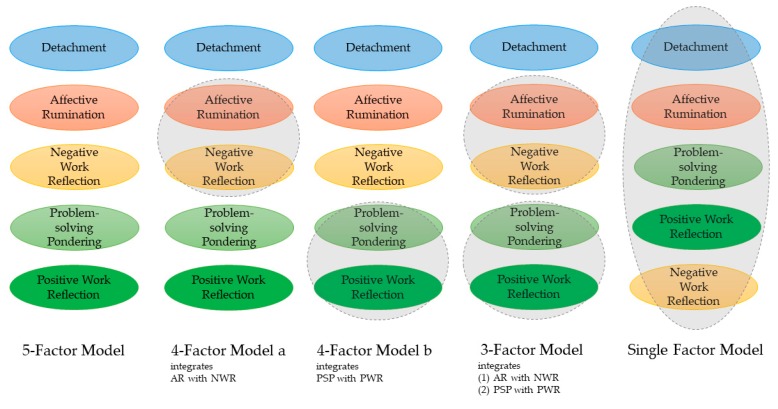
Focal 5-factor model and alternative models for examination of the factorial structure of work-related rumination.

**Figure 2 ijerph-16-02987-f002:**
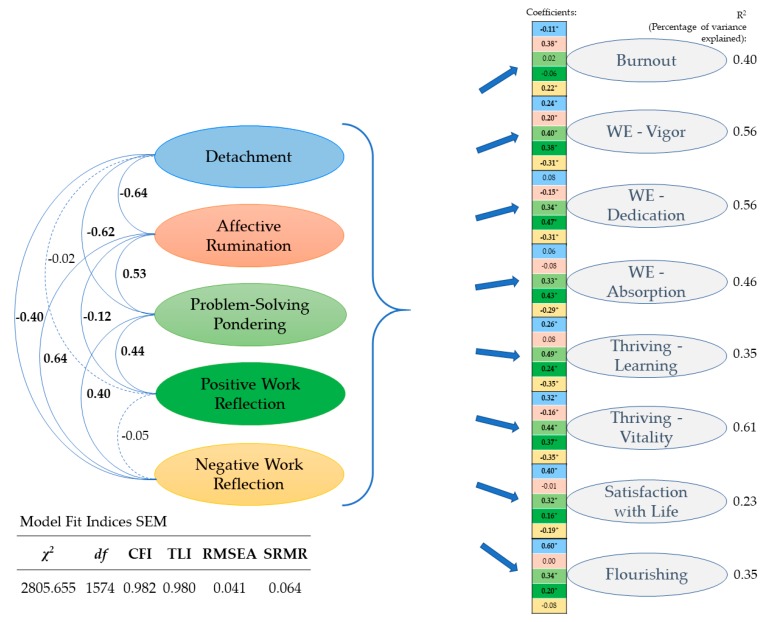
Focal structural equation model linking facets of work-related rumination to facets of employee well-being.

**Figure 3 ijerph-16-02987-f003:**
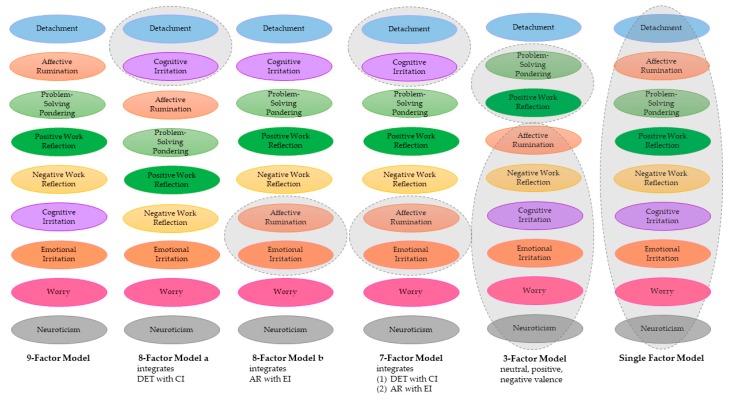
Focal 9-factor model and alternative models for examination of the factorial structure of work-related rumination and its concomitants.

**Table 1 ijerph-16-02987-t001:** Means, standard deviations, and zero-order correlations between study variables.

	Variable	M	SD	1	2	3	4	5	6	7	8	9	10	11	12	13	14	15	16	17	18	19	20	21	22	23
1.	Sex	1.72	0.45	-																						
2.	Age	34.34	9.55	0.01	-																					
3.	Working Time/Week	28.93	11.26	−0.17 **	0.32 **	-																				
4.	Detachment	3.47	0.90	−0.02	0.04	−0.12 **	(0.85)																			
5.	Affective Rumination	2.42	0.91	0.03	−0.06	0.12 **	−0.56 **	(0.90)																		
6.	PSP	2.66	0.79	0.03	0.03	0.09	−0.52 **	0.42 **	(0.82)																	
7.	PWR	2.94	0.87	0.13 **	−0.04	−0.11 *	−0.02	−0.11 *	0.40 **	(0.92)																
8.	NWR	2.80	0.95	0.04	−0.10 *	−0.03	−0.36 **	0.58 **	0.31 **	−0.04	(0.91)															
9.	Burnout	2.36	0.97	0.08	−0.12 *	−0.01	−0.40 **	0.55 **	0.29 **	−0.09 *	0.47 **	(0.91)														
10.	WE—Vigor	3.61	1.32	0.06	−0.05	−0.06	0.19 **	−0.34 **	0.19 **	0.53 **	−0.35 **	−0.31 **	(0.86)													
11.	WE—Dedication	3.70	1.27	0.08	−0.01	−0.01	0.06	−0.25 **	0.29 **	0.60 **	−0.30 **	−0.24 **	0.83 **	(0.91)												
12.	WE—Absorption	3.66	1.49	0.08	−0.03	−0.01	0.00	−0.16 **	0.31 **	0.55 **	−0.23 **	−0.16 **	0.79 **	0.87 **	(0.89)											
13.	Work Engagement	3.46	1.45	0.08	−0.03	−0.03	0.09	−0.26 **	0.28 **	0.60 **	−0.31 **	−0.25 **	0.92 **	0.96 **	0.95 **	(0.95)										
14.	Thriving—Learning	4.75	1.34	−0.04	0.06	0.08	0.03	−0.09	0.29 **	0.41 **	−0.22 **	−0.13 **	0.54 **	0.61 **	0.53 **	0.60 **	(0.91)									
15.	Thriving—Vitality	4.22	1.23	0.04	0.02	−0.04	0.24 **	−0.39 **	0.15 **	0.52 **	−0.42 **	−0.40 **	0.82 **	0.77 **	0.70 **	0.80 **	0.66 **	(0.85)								
16.	Thriving	4.48	1.17	0.00	0.05	0.02	0.15 **	−0.25 **	0.24 **	0.51 **	−0.35 **	−0.28 **	0.74 **	0.76 **	0.67 **	0.76 **	0.92 **	0.90 **	(0.80)							
17.	Satisfaction with Life	4.63	1.28	0.00	0.09 *	0.01	0.23 **	−0.21 **	0.08	0.28 **	−0.21 **	−0.29 **	0.48 **	0.44 **	0.37 **	0.45 **	0.32 **	0.42 **	0.40 **	(0.91)						
18.	Flourishing	5.23	1.11	0.05	0.10 *	−0.03	0.38 **	−0.26 **	0.02	0.31 **	−0.18 **	−0.29 **	0.49 **	0.44 **	0.35 **	0.45 **	0.38 **	0.48 **	0.47 **	0.77 **	(0.93)					
19.	Cognitive Irritation	2.43	0.95	0.07	0.03	0.18 **	−0.77 **	0.69 **	0.66 **	0.12 *	0.49 **	0.50 **	−0.12 *	0.01	0.07	−0.01	0.07	−0.17 **	−0.05	−0.12 *	−0.21 **	(0.88)				
20.	Emotional Irritation	2.16	0.83	0.03	−0.06	0.05	−0.49 **	0.67 **	0.35 **	−0.07	0.55 **	0.56 **	−0.32 **	−0.22 **	−0.13 **	−0.23 **	−0.20 **	−0.39 **	−0.32 **	−0.34 **	−0.41 **	0.61 **	(0.89)			
21.	Irritation	2.26	0.79	0.05	−0.03	0.12*	−0.69 **	0.75 **	0.53 **	0.00	0.58 **	0.60 **	−0.27 **	−0.14 **	−0.05	−0.16 **	−0.10 *	−0.33 **	−0.23 **	−0.28 **	−0.36 **	0.85 **	0.93 **	(0.90)		
22.	Worry	2.90	0.75	0.18 **	−0.14 **	−0.12 **	−0.41 **	0.49**	0.32 **	0.04	0.51 **	0.55 **	−0.27 **	−0.18 **	−0.12 **	−0.20 **	−0.11 *	−0.29 **	−0.22 **	−0.35 **	−0.31 **	0.47 **	0.58 **	0.59**	(0.93)	
23.	Neuroticism	3.08	1.03	0.02	−0.15 **	−0.05	−0.30 **	0.40 **	0.14 **	−0.05	0.39 **	0.43 **	−0.29 **	−0.23 **	−0.15 **	−0.24 **	−0.15 **	−0.29 **	−0.24 **	−0.39 **	−0.41 **	0.34 **	0.63 **	0.56**	0.54**	(0.85)

Note. Cronbach’s α reliabilities are placed on the diagonal in parentheses. *****
*p* < 0.05. ******
*p* < 0.01. NWR = Negative Work Reflection; PWR = Positive Work Reflection; PSP = Problem-Solving Pondering; WE = Work Engagement.

**Table 2 ijerph-16-02987-t002:** Fit indices and *χ*^2^-difference test among alternative measurement models of the factorial structure of work-related rumination.

Model	*χ* ^2^	*df*	CFI	TLI	RMSEA	SRMR	Δ*df*	Δ*χ*^2^	Preference
Single Factor Model	2159.2	189	0.819	0.798	0.148	0.162	1	1555.32	5-Factor Model
3-Factor Model (AR + NWR, PSP + PWR)	1562.5	186	0.873	0.857	0.125	0.138	7	958.603	5-Factor Model
4-Factor Model b (PSP + PWR)	1397.4	183	0.888	0.872	0.118	0.131	4	793.456	5-Factor Model
4-Factor Model a (AR + NWR)	771.9	183	0.946	0.938	0.082	0.093	4	167.97	5-Factor Model
5 Factor Model	603.93	179	0.961	0.954	0.071	0.082			

Note: Δ*df* and Δ*χ*^2^ refer to the comparisons with the best fitting model. AR = Affective Rumination; CFI = Comparative Fit Index; NWR = Negative Work Reflection; PSP = Problem-Solving Pondering; PWR = Positive Work Reflection; RMSEA = Root Mean Square Error of Approximation; SRMR = Standardized Root Mean Square Residual; TLI = Tucker–Lewis Index.

**Table 3 ijerph-16-02987-t003:** Recommendation for fit indices.

Fit Index	Good Fit	Acceptable Fit
*χ* ^2^	0 ≤ *χ*^2^ ≤ 2*df*	2*df* < *χ*^2^ ≤ 3*df*
*p*-value	0.05 < *p* ≤ 1.00	0.01 ≤ *p* ≤ 0.05
RMSEA	0 ≤ RMSEA ≤ 0.05	0.05 < RMSEA ≤ 0.08
SRMR	0 ≤ SRMR ≤ 0.05	0.05 < SRMR ≤ 0.10
TLI	0.97 ≤ TLI ≤ 1.00	0.95 ≤ TLI < 0.97
CFI	0.97 ≤ CFI ≤ 1.00	0.95 ≤ CFI < 0.97
additional recommended for nested models:		
*χ^2^*-Difference Test		

Note. RMSEA = Root Mean Square Error of Approximation; SMSR = Standardized Root Mean Square Residual; TLI = Tucker–Lewis Index; CFI = Comparative Fit Index.

**Table 4 ijerph-16-02987-t004:** Fit indices and *χ*^2^-difference test among alternative measurement models of work-related rumination and their concomitants.

Model	*χ* ^2^	*df*	CFI	TLI	RMSEA	SRMR	Δ*df*	Δ*ꭓ*^2^	Preference
Single Factor Model	7408.965	1325	0.900	0.896	0.099	0.117	36	5164.308	9-Factor Model
3-Factor Model (neutral, positive, negative)	6589.688	1322	0.913	0.910	0.092	0.110	33	4345.031	9-Factor Model
7-Factor Model (AR + EI, DET + CI)	2641.922	1304	0.978	0.977	0.047	0.069	15	397.265	9-Factor Model
8-Factor Model b (AR + EI)	2576.69	1297	0.979	0.978	0.046	0.068	8	332.032	9-Factor Model
8-Factor Model a (DET + CI)	2309.257	1297	0.983	0.982	0.041	0.065	8	64.6	9-Factor Model
9-Factor Model	2244.657	1289	0.984	0.983	0.040	0.063			

Note. Δ*df* and Δ*χ*^2^ refer to comparisons with the best fitting model. AR = Affective Rumination; CFI = Comparative Fit Index; DET = Detachment; EI = Emotional Irritation; CI = Cognitive Irritation; RMSEA = Root Mean Square Error of Approximation; SRMR = Standardized Root Mean Square Residual; TLI = Tucker–Lewis Index.

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
