# Peer review of "My Mind is Working Overtime—Towards an Integrative Perspective of Psychological Detachment, Work-Related Rumination, and Work Reflection"

_ijerph, 2019, doi:10.3390/ijerph16162987_

Round 1

Reviewer 1 Report

I read the manuscript with great interest. The paper is informative but there are several concerns that need to address or reconsider. Please see my comments and suggestions below.

The authors believe that the five aspects of thinking about work overlap with each other and thus an integrated comparison is needed. Although benefits of the five concepts examinations were explained, why those five concepts were selected (hopefully not cherry-picking) and why the comparison is important was not clearly stated in the paper.

From my understanding, positive and negative work reflection is not comparable to the other three because the former can occur any time while the latter are concepts of doing after work.

Each variable in this paper was developed to construct a specific meaning. Differences and similarities among the variables (five main plus eight additional) are obvious in light of their meanings. Thus, I am skeptical that hypotheses are worth it. It will be rather very surprising if any variables are nested in the same factor without theoretical support. The authors may consider different research designs such as concurrent/discriminant validity, predictive validity, and multiple regression. 

It is better to look at correlation coefficients when comparing relationships of different variables than using a confirmative factor analysis. 

Author Response

Comments and Suggestions for Authors

I read the manuscript with great interest. The paper is informative but there are several concerns that need to address or reconsider. Please see my comments and suggestions below.

Our response:

Thank you for your constructive comments on our manuscript.

The authors believe that the five aspects of thinking about work overlap with each other and thus an integrated comparison is needed. Although benefits of the five concepts examinations were explained, why those five concepts were selected (hopefully not cherry-picking) and why the comparison is important was not clearly stated in the paper.

Our response:

Thank you for pointing out that the choice of constructs/facets of rumination should be justified more explicitly. We have reviewed the literature on work-related rumination. From our literature search psychological detachment, affective rumination, problem-solving pondering, positive work reflection, and negative work reflection are the five major aspects (and scales) studied in mainstream (top-tier) industrial and organizational psychology and occupational health psychology journals. We supplemented our analysis by concomitants of work-related rumination like, for instance, irritation. Our focus is also in line with recent reviews and meta-analyses on psychological detachment and recovery. Our manuscript is aimed at informing researchers interested in the established facets of work-related rumination. We believe that the list of facets is not arbitrary but exhaustive (regarding work-related rumination), although some authors may use varying labels for essentially similar aspects of work-related rumination (Frone, 2015). If there is a large volume on work-related rumination outside of the major I/O-journals which taps into unique aspects beyond the major facets and would supplement our discussion, please point us to this work. We will be happy to include these aspects in the discussion section.

Comparison of the five facets is important because they have been studied in a largely isolated way so far. Authors from different labs also tend to use similar arguments regarding rumination but apply different operationalizations/measures of rumination. At this point, we see a need for clarification whether the different facets/scales measure something different or whether essentially the same phenomenon is given just different labels by different people (e.g., affective rumination vs. negative work reflection). Beyond merely distinguishing facets gaining a comprehensive picture of the associations among the facets is important when having to decide for or against a specific operationalization. We have double-checked our manuscript to make sure that we convey this basic idea clearly. In the introduction section on page 2 we have added a few sentences on the importance of considering differential association or discriminant/convergent validity.

From my understanding, positive and negative work reflection is not comparable to the other three because the former can occur any time while the latter are concepts of doing after work.

Our response:

We appreciate your valuable comment on the conceptual distinction between work reflection and the other facets considered. In line with prior research on work reflection, we asked participants to refer to *thoughts about work in off-job time*. Hence, we believe that work reflection *as measured in your study* refers to the same time frame and set of opportunities for thinking about work as the other facets of rumination. In response to your comment, we have double-checked the description of the reflection scale in the methods section:

We measured positive work reflection using the 3-item-scale developed by Fritz and Sonnentag (Fritz & Sonnentag, 2005). The items referred to how often positive thoughts about work occur during leisure. A sample item is “During leisure time, I consider the positive aspects of my job.”.

We measured negative work reflection using the 3-item-scale developed by Fritz and Sonnentag (Fritz & Sonnentag, 2006) The items referred to how often negative thoughts about the job arise after work. A sample item is “During leisure time, I consider the negative aspects of my job.”.

In our view, the description of the scales are explicit and clear. Our instructions are in line with the standard measures of positive and negative work reflection as conceptualized by the initial authors of the scales. Hence, in our view, the scales of work reflection are more similar regarding the scope and time frame than one might expect. Of note, the considerable correlations between, for instance, negative work reflection and psychological detachment or affective rumination suggest that reflection and the other facets of work-related rumination do covary to a considerable extent and may not be dissimilar in conceptual or measurement terms at all.

Each variable in this paper was developed to construct a specific meaning. Differences and similarities among the variables (five main plus eight additional) are obvious in light of their meanings. Thus, I am skeptical that hypotheses are worth it. It will be rather very surprising if any variables are nested in the same factor without theoretical support. The authors may consider different research designs such as concurrent/discriminant validity, predictive validity, and multiple regression. 

Our response:

We appreciate your comments regarding the limitations of the methods applied and the analytic strategy used. Our study was aimed at examining convergent and discriminant validity. More specifically, we investigate whether the five facets of work-related rumination yield differential associations with a set of eight indicators of work-related and general well-being. In our view, this analysis clearly taps into (our understanding of) convergent (We think, you meant *convergent* validity rather than *concurrent* validity) vs. discriminant validity. Predictor A highly relates to criterion C but is unrelated to criterion D, while predictor B is unrelated to criterion C, but strongly relates to criterion D. In fact, we find that for instance psychological detachment is highly related to satisfaction with life and flourishing, but does not explain variance in facets of work engagement. At the same time affective rumination is linked to burnout, but does not explain much variance in satisfaction with life and flourishing. In our view, these results provide evidence for the discriminant validity of the five facets of work-related rumination. In a similar vein, we found that particularly positive facets of work-related rumination consistently relate to energetic states. This finding provides evidence for convergent validity. Accordingly, in the revised version of the manuscript, we have taken care to point out more clearly that our findings inform issues of convergent vs. discriminant validity. For instance in the introduction section on page 2 and in the discussion section on page 16 we highlight that our study provides insights on the convergent vs. discriminant validity of the five facets of work-related rumination.

From our point of view, obtaining five distinct factors of work-related rumination is less trivial than it may appear at first sight. The very high correlations between cognitive irritation and lack of psychological detachment provide evidence that there are scales in the literature, which are barely distinct (the 8-factor model doesn’t fit much worse than the 9-factor model, albeit the Chi-square difference test is significant), although the conceptual foundation for deriving these scales is different. In the literature on work-related rumination some authors tend to refer to some of the five aspects of rumination at times interchangeably to argue for their specific hypotheses. In this sense, the distinction is not as clear cut as it may appear. Empirical evidence regarding this issue helps to go beyond theoretical speculation, to what extent the facets are independent and to what extent they are related. In our view, there is much value in providing empirical evidence for *how closely* the facets are related among each other to gain a clearer picture of the structure of work-related rumination – a question that cannot be addressed sufficiently by conceptual arguments alone.

We agree that examination of predictive validity (using time lagged criteria) would have been advantageous. From a practical point of view it is at times challenging to collect data on a broad set of scales and at the same time to obtain time lagged or even longitudinal data for all these (often quite long) sets of items. In our study, we have explicitly focused on concurrently measuring a broad set of variables to make sure, measurement is reliable and valid. Our focus was examining the structure of work-related rumination and differential associations with a comprehensive set of well-being indicators. The breadth of concepts is a major strength of this study, which we would have to give up in a longitudinal study. Although, the bivariate links between work-related rumination and well-being may be inflated due to common method bias, the results on the differential links to well-being (convergent and discriminant validity) are not trivial and cannot be explained easily by methodological artifacts. Please note that predicting time lagged well-being might even provide less clear conclusions, if the five facets of rumination differ regarding the speed in which they affect a given indicator of employee well-being. If, for instance, affective rumination yields immediate effects, but lack of detachment affects well-being at a slower speed and in the long run only, comparing the effect of affective rumination vs. psychological detachment on burnout at t2 (e.g. two months or one year later) would be ambiguous (see for instance (Dormann & Griffin, 2015; George & Jones, 2000; Mitchell & James, 2001). To cut a long story short, to make sure your comment is reflected in the paper, we included a short paragraph on page 18. More specifically, in the future research section, we propose considering the (differential) predictive validity of the five facets of work-related rumination and considering issues of timing.

We leverage structural equation modeling (SEM) to scrutinize the associations between work-related rumination and employee well-being. In our view, applying (latent variable) structural equation modeling is superior to using linear regression. We think, that most authors of textbooks on SEM would agree. For one, measurement error is taken into account. For the other, some of the scales violate assumptions necessary for multiple regression analysis (multivariate normality assumption) and our SEM account for these problems by applying diagonally-weighted least squares estimation. Finally, SEM allows for testing all coefficients in one single model rather than running eight separate regression models, an option that is always preferred by scholars taking research methods seriously. On page 10 we explain the advantages of SEM. Readers interested in the strength of the bivariate correlations may consult the correlation table in our manuscript.

It is better to look at correlation coefficients when comparing relationships of different variables than using a confirmative factor analysis. 

Our response:

Thank you for sharing your view with us. Our rationale for focusing on the confirmatory factor analysis (CFA) was that applying SEM allows taking measurement error into account. Hence, the estimated standardized covariances may represent the *true association* more accurately than the correlations among the scale means (in which each item, no matter how reliable it is, is weighted equally). Our focal models also take into account that some scales or items may not be normally distributed etc. In our view, there are good reasons for focusing on the standardized covariances from the CFA. However, please note that we *do* report all the raw correlations in the correlation table. We believe this is an elegant way to make everyone happy: Those how prefer the SEM coefficients and those who like to take a closer look at the raw correlations. In the case of our study, conclusions drawn from both perspectives (high/moderate/small/no association) do not seem to differ dramatically. We hope that our arguments make sense to you and will agree with us that our manuscript provides “the best of both worlds”.

References

Dormann, C., & Griffin, M. A. (2015). Optimal time lags in panel studies. Psychological Methods. https://doi.org/10.1037/met0000041

Fritz, C., & Sonnentag, S. (2005). Recovery, health, and job performance: Effects of weekend experiences. Journal of Occupational Health Psychology, 10(3), 187–199. https://doi.org/10.1037/1076-8998.10.3.187

Fritz, C., & Sonnentag, S. (2006). Recovery, well-being, and performance-related outcomes: The role of workload and vacation experiences. Journal of Applied Psychology, 91(4), 936–945. https://doi.org/10.1037/0021-9010.91.4.936

Frone, M. R. (2015). Relations of negative and positive work experiences to employee alcohol use: Testing the intervening role of negative and positive work rumination. Journal of Occupational Health Psychology, 20(2), 148–160. https://doi.org/10.1037/a0038375

George, J. M., & Jones, G. R. (2000). The role of time in theory and theory building. Journal of Management, 26(4), 657–684. https://doi.org/10.1177/014920630002600404

Mitchell, T. R., & James, L. R. (2001). Building better theory: Time and the specification of when things happen. Academy of Management Review, 26(4), 530–547. https://doi.org/10.2307/3560240

Reviewer 2 Report

The manuscript describes the hypothesis, study, and methods of an original research. The authors tackle an original and innovative topic, responding to the need for new knowledge in the sector, especially in Europe.

The English language is discreet. However, there are several inaccuracies both orthographic and grammatical. Therefore, a revision of the manuscript from this point of view is required.

Abstract is just enough since the description of your study's objectives is too distracting. You must be clearer and more concise. Keywords are discreet both in terms of appropriateness of context and the purpose of study.

The introduction is sufficiently well written and satisfactory in terms of appropriateness of context and the purpose of study. The analysis of the literature must be improved considering, for example, the following researches:

Banerjee M et al. Barriers to Mindfulness: a Path Analytic Model Exploring the Role of Rumination and Worry in Predicting Psychological and Physical Engagement in an Online Mindfulness-Based Intervention. Mindfulness (N Y). 2018;9(3):980-992.

Ebert DD, et al. Restoring depleted resources: Efficacy and mechanisms of change of an internet-based unguided recovery training for better sleep and psychological detachment from work. Health Psychol. 2015 Dec;34S:1240-1251. doi: 10.1037/hea0000277.

Sonnentag S, Schiffner C. Psychological Detachment from Work during Nonwork Time and Employee Well-Being: The Role of Leader's Detachment. Span J Psychol. 2019 Mar 1;22:E3. doi: 10.1017/sjp.2019.2.

Vandevala T. Psychological rumination and recovery from work in intensive care professionals: associations with stress, burnout, depression and health. J Intensive Care. 2017 Feb 2;5:16. doi: 10.1186/s40560-017-0209-0.

Wang X, et al. The Relationship Between Psychological Detachment and Employee Well-Being: The Mediating Effect of Self-Discrepant Time Allocation at Work. Front Psychol. 2018 Dec 11;9:2426. doi: 10.3389/fpsyg.2018.02426.

I also recommend you to consider a brief discussion regarding the potential additional effects of the economic crisis in the dynamics of occupational engagement. For example, can a global crisis context interfere with engagement, psychological detachment, or affective rumination? I state that there are no specific references to literature and, especially for this, it would be interesting that you reflect on this aspect. Regarding the dynamics of the economic crisis, you can refer to the following publications:

Giorgi G et al. Economic stress in workplace: The impact of fear the crisis on mental health. Work 2015; 51(1): 135-142. doi: 10.3233/WOR-141844.

Mucci N et al. The correlation between stress and economic crisis: a systematic review. Neuropsychiatr Dis Treat 2016; 12:983-993. doi: 10.2147/NDT.S98525.

Methods section appears of a discreet quality. Add also any possible consideration about ethics statement in the Methods body text. The statistical methodologies that you have used are sufficiently well illustrated.

Results section is, as a whole, of a satisfactory quality. However, you should better connect the text with the 4 tables and the 3 figures. In other words, you should more thoroughly explain the contents of the tables and the figures before referring to them in the body text. The graphic quality of the figures is very good.

The discussion section is overall sufficient. You should more carefully compare your findings with the field literature. I suggest you to go deeper into the study's limitations. Finally, I suggest you to use a final paragraph to sum up your findings. Otherwise, you should also carefully explain what is the specific contribution that your findings bring to literature and knowledge in this area.

Check accurately all the quotes in the brackets in the text and in the Reference section. These must strictly comply with the Author's guidelines.

Author Response

Comments and Suggestions for Authors

The manuscript describes the hypothesis, study, and methods of an original research. The authors tackle an original and innovative topic, responding to the need for new knowledge in the sector, especially in Europe.

Our response:

Thank you for your positive general assessment. We are happy that you have provided a balanced assessment of our manuscript and explicitly mention lots of positive aspects. We appreciate your valuable suggestions and we have tried to make sure all of them are reflected in the revised version of the paper.

The English language is discreet. However, there are several inaccuracies both orthographic and grammatical. Therefore, a revision of the manuscript from this point of view is required.

Our response:

Thank you for your positive feedback. We have consulted a native speaker for improving language and style of the revised version of the manuscript. All changes are set in red font. 

Abstract is just enough since the description of your study's objectives is too distracting. You must be clearer and more concise. Keywords are discreet both in terms of appropriateness of context and the purpose of study.

 Our response:

Thank you for your positive feedback. We have tried to revise the abstract according to your comments. We were not quite sure *exactly which points* need clarification and exactly which aspects should be presented more concisely. Presenting more concisely often implies including less details (and being less specific/clear as well). We have added one element and now explicitly state the strongest associations between work-related rumination and the well-being facets. The pictorial abstract/Figure 2 summarizes the essence of the study to the readers as well. Although, we have left most parts of the abstract unchanged, we hope that the revised abstract is more in line with the standards and with what you expect from authors for this journal. In our view, the abstract is meant to communicate the essential results and make reader curious to read the paper. We hope that the revised abstract complies with this mission.

The introduction is sufficiently well written and satisfactory in terms of appropriateness of context and the purpose of study. The analysis of the literature must be improved considering, for example, the following researches:

Banerjee M et al. Barriers to Mindfulness: a Path Analytic Model Exploring the Role of Rumination and Worry in Predicting Psychological and Physical Engagement in an Online Mindfulness-Based Intervention. Mindfulness (N Y). 2018;9(3):980-992. Ebert DD, et al. Restoring depleted resources: Efficacy and mechanisms of change of an internet-based unguided recovery training for better sleep and psychological detachment from work. Health Psychol. 2015 Dec;34S:1240-1251. doi: 10.1037/hea0000277. Sonnentag S, Schiffner C. Psychological Detachment from Work during Nonwork Time and Employee Well-Being: The Role of Leader's Detachment. Span J Psychol. 2019 Mar 1;22:E3. doi: 10.1017/sjp.2019.2. Vandevala T. Psychological rumination and recovery from work in intensive care professionals: associations with stress, burnout, depression and health. J Intensive Care. 2017 Feb 2;5:16. doi: 10.1186/s40560-017-0209-0. Wang X, et al. The Relationship Between Psychological Detachment and Employee Well-Being: The Mediating Effect of Self-Discrepant Time Allocation at Work. Front Psychol. 2018 Dec 11;9:2426. doi: 10.3389/fpsyg.2018.02426.

 Our response:

Thank you for pointing us to a set of studies to supplement our review of the literature. We believe that these papers are very helpful to strengthen the rationale particularly with regard to associations with well-being outcomes. We have taken care to include these studies. More specifically, we  have embedded all the papers in our review. Please note that the Banerjee et al.-study does not refer to *work* engagement. Hence, we only included it to provide evidence for the rumination-worry link. We hope that we are able to embed the papers in a way that is in line with what you had in mind. Thank you again for your insightful suggestions to leverage these papers for our line of argument.

I also recommend you to consider a brief discussion regarding the potential additional effects of the economic crisis in the dynamics of occupational engagement. For example, can a global crisis context interfere with engagement, psychological detachment, or affective rumination? I state that there are no specific references to literature and, especially for this, it would be interesting that you reflect on this aspect. Regarding the dynamics of the economic crisis, you can refer to the following publications:

Giorgi G et al. Economic stress in workplace: The impact of fear the crisis on mental health. Work 2015; 51(1): 135-142. doi: 10.3233/WOR-141844. Mucci N et al. The correlation between stress and economic crisis: a systematic review. Neuropsychiatr Dis Treat 2016; 12:983-993. doi: 10.2147/NDT.S98525.

Our response:

Thank you for pointing us to this very interesting issue. We have followed your advice to reflect on economic context and how it may relate to the focal variables in our study. In essence, we are basically interested in the *relations* among the focal variables rather than the *level* of these variables. However, we understand that the economic context may differentially affect the focal variables and hence change the patterns of associations among these variables. In the future research section on page 21,we have added one paragraph on this issue. Given the considerable length of the revised manuscript, we have tried to keep this issue rather short. We hope our reflection taps into what you had in mind when proposing the issue.

Methods section appears of a discreet quality. Add also any possible consideration about ethics statement in the Methods body text. The statistical methodologies that you have used are sufficiently well illustrated.

Our response:

Thank you for reminding us to mention that our study was in line with ethical standards. We have included an additional paragraph in the procedure section on page 8. In explicitly state that our study complied with the ethical standards of our science.

Results section is, as a whole, of a satisfactory quality. However, you should better connect the text with the 4 tables and the 3 figures. In other words, you should more thoroughly explain the contents of the tables and the figures before referring to them in the body text. The graphic quality of the figures is very good.

Our response:

Thank you for encouraging more elaboration on the interpretation of the coefficients, tables, and figures. We have included some additional explanation throughout the results section. We hope that our additional elaboration of the tables and figures contribute to embed these elements into the results section and the manuscript as whole. We are also confident that the results section is more reader friendly, thanks to your suggestion.

The discussion section is overall sufficient. You should more carefully compare your findings with the field literature. I suggest you to go deeper into the study's limitations. Finally, I suggest you to use a final paragraph to sum up your findings. Otherwise, you should also carefully explain what is the specific contribution that your findings bring to literature and knowledge in this area.

 Our response:

Thank you pointing out that we should pay attention to reflect on our findings in the light of prior evidence. In the revised version of the manuscript we explicitly relate our findings to evidence from prior research in the theoretical implications section. All additions are marked in red font.

Furthermore, in response to your suggestion, we have extended the limitations section. More specifically, on page 20 we have elaborated more on the limitations and have added, for instance,  that we might have omitted a couple of relevant variables. However, we also felt a strong need to point out that for instance the design of our study is suited to address the focal aim (increasing conceptual clarity regarding work-related rumination). We hope, the additional reflection is in line with what you had in mind with your comment.

In line with your suggestion, we have included a final paragraph on the specific contribution of our paper to the literature on page 19. We hope that the paragraph is non-redundant with other paragraphs in our manuscript and we hope that it actually clarifies what we think are the basic contributions to the literature.

Check accurately all the quotes in the brackets in the text and in the Reference section. These must strictly comply with the Author's guidelines.

Our response:

We have double-checked that page numbers are presented for each and every literal quote. Furthermore, we have double-checked that we use the zotero style template of the journal and that the references section is aligned with the MDPI-guidelines.

Round 2

Reviewer 1 Report

I appreciate your detailed responses to my comments and efforts to improve your manuscript. Now, I better understand your intentions and believe that the manuscript is much better. 

Regarding the selection of the five concepts, you explained how you found them in your response but it is necessary to applied to the introduction of your paper as well. Since those five consist of the based framework, more information about them may be needed. 

You can add your points on validity tests in your response (the last paragraph of page 2) as another problem of why your study is necessary. 

In the strengths and limitations section, authors stated the limitation of self-reported data from a single source. Authors may run a common method variance test (Harman's is not recommended) to check the common method bias and can provide an appropriate remedy if much bias is found.     

Reviewer 2 Report

Only a careful re-reading to fix few linguistic inaccuracies Is required.
